# Learning Hierarchical Priors in VAEs

**Alexej Klushyn**[1][2]  **Nutan Chen**[1]  **Richard Kurle**[1][2]  **Botond Cseke**[1]  **Patrick van der Smagt**[1]

[1]Machine Learning Research Lab, Volkswagen Group, Germany
[2]Department of Informatics, Technical University of Munich, Germany
`{alexej.klushyn, nutan.chen, richardk, botond.cseke, smagt}@argmax.ai`

## Abstract

We propose to learn a hierarchical prior in the context of variational autoencoders to avoid the over-regularisation resulting from a standard normal prior distribution. To incentivise an informative latent representation of the data, we formulate the learning problem as a constrained optimisation problem by extending the *Taming VAEs* framework to two-level hierarchical models. We introduce a graph-based interpolation method, which shows that the topology of the learned latent representation corresponds to the topology of the data manifold—and present several examples, where desired properties of latent representation such as smoothness and simple explanatory factors are learned by the prior.

## 1 Introduction

Variational autoencoders (VAEs) [15, 24] are a class of probabilistic latent variable models for unsupervised learning. The learned generative model and the corresponding (approximate) posterior distribution of the latent variables provide a decoder/encoder pair that often captures semantically meaningful features of the data. In this paper, we address the issue of learning informative latent representations/encodings of the data.

The vanilla VAE uses a standard normal prior distribution for the latent variables. It has been shown that this can lead to over-regularising the posterior distribution, resulting in latent representations that do not represent well the structure of the data [1]. There are several approaches to alleviate this problem: (i) defining and learning complex prior distributions that can better model the encoded data manifold [10, 28]; (ii) using specialised optimisation algorithms, which try to find local/global minima of the training objective that correspond to informative latent representations [4, 27, 14, 25]; and (iii) adding mutual-information-based constraints or regularisers to incentivise a good correspondence between the data and the latent variables [1, 31, 9]. In this paper, we focus on the first two approaches.

We use a two-level stochastic model, where the first layer corresponds to the latent representation and the second layer models a hierarchical prior (continuous mixture). In order to learn such hierarchical priors, we extend the optimisation framework introduced in [25], where the authors reformulate the VAE objective as the Lagrangian of a constrained optimisation problem. They impose an inequality constraint on the reconstruction error and use the KL divergence between the approximate posterior and the standard normal prior as the optimisation objective. We substitute the standard normal prior with the hierarchical one and use an importance-weighted bound [5] to approximate the resulting intractable marginal. Concurrently, we introduce the associated optimisation algorithm, which is inspired by GECO [25]—the latter does not always lead to good encodings (e.g., Sec. 4.1). Our approach better avoids posterior collapse and enhances interpretability compared to similar methods.

We adopt the manifold hypothesis [6, 26] to validate the quality of a latent representation. We do this by proposing a nearest-neighbour graph-based method for interpolating between different data points along the learned data manifold in the latent space.

## 2   Methods

### 2.1   VAEs as a Constrained Optimisation Problem

VAEs model the distribution of i.i.d. data $\mathcal{D} = \{\mathbf{x}_i\}_{i=1}^N$ as the marginal

$$\prod_i p_\theta(\mathbf{x}_i) = \prod_i \int p_\theta(\mathbf{x}_i|\mathbf{z})\, p(\mathbf{z})\, \mathrm{d}\mathbf{z}. \tag{1}$$

The model parameters are learned through amortised variational EM, which requires learning an approximate posterior distribution $q_\phi(\mathbf{z}|\mathbf{x}_i) \approx p_\theta(\mathbf{z}|\mathbf{x}_i)$. It is hoped that the learned $q_\phi(\mathbf{z}|\mathbf{x})$ and $p_\theta(\mathbf{x}|\mathbf{z})$ result in an informative latent representation of the data. For example, $\{\mathbb{E}_{q_\theta(\mathbf{z}|\mathbf{x}_i)}[\mathbf{z}]\}_{i=1}^N$ cluster w.r.t. some discrete features or important factors of variation in the data. In Sec. 4.1, we show a toy example, where the model can learn the true underlying factors of variation in $\mathcal{D}$.

Amortised variational EM in VAEs maximises the evidence lower bound (ELBO) [15, 24]:

$$\mathbb{E}_{p_\mathcal{D}(\mathbf{x})}\big[\log p_\theta(\mathbf{x})\big] \geq \mathcal{F}_{\text{ELBO}}(\theta,\phi) \equiv \mathbb{E}_{p_\mathcal{D}(\mathbf{x})}\Big[\mathbb{E}_{q_\phi(\mathbf{z}|\mathbf{x})}\big[\log p_\theta(\mathbf{x}|\mathbf{z})\big] - \mathbb{KL}\big(q_\phi(\mathbf{z}|\mathbf{x})\|\, p(\mathbf{z})\big)\Big], \tag{2}$$

where $q_\phi(\mathbf{z}|\mathbf{x})$ and $p_\theta(\mathbf{x}|\mathbf{z})$ are typically assumed to be diagonal Gaussians with their parameters defined as neural network functions of the conditioning variables. $p_\mathcal{D}(\mathbf{x}) = \frac{1}{N}\sum_{i=1}^N \delta(\mathbf{x} - \mathbf{x}_i)$ stands for the empirical distribution of $\mathcal{D}$. The (EM) optimisation problem [e.g. 21] is formulated as

$$\min_{\theta,\phi} -\mathcal{F}_{\text{ELBO}}(\theta,\phi) \,\, \widehat{=}\,\, \min_\theta \min_\phi -\mathcal{F}_{\text{ELBO}}(\theta,\phi). \tag{3}$$

The corresponding optimisation algorithm was originally introduced as a double-loop algorithm, however, in the context of VAEs—or neural inference models in general—it is a common practice to optimise $(\theta,\phi)$ jointly.

It has been shown that local minima with high ELBO values do not necessarily result in informative latent representations [1, 14]. In order to address this problem, several approaches have been developed, which typically result in some weighting schedule for either the negative expected log-likelihood or the KL term of the ELBO [4, 27]. This is because a different ratio targets different regions in the rate-distortion plane, either favouring better compression or reconstruction [1].

In [25], the authors reformulate the VAE objective as the Lagrangian of a constrained optimisation problem. They choose $\mathbb{KL}\big(q_\phi(\mathbf{z}|\mathbf{x})\|\, p(\mathbf{z})\big)$ as the optimisation objective and impose the inequality constraint $\mathbb{E}_{p_\mathcal{D}(\mathbf{x})}\mathbb{E}_{q_\phi(\mathbf{z}|\mathbf{x})}\big[\mathrm{C}_\theta(\mathbf{x},\mathbf{z})\big] \leq \kappa^2$. Typically $\mathrm{C}_\theta(\mathbf{x},\mathbf{z})$ is defined as the reconstruction-error-related term in $-\log p_\theta(\mathbf{x}|\mathbf{z})$. Since $\mathbb{E}_{p_\mathcal{D}(\mathbf{x})}\mathbb{E}_{q_\phi(\mathbf{z}|\mathbf{x})}\big[\mathrm{C}_\theta(\mathbf{x},\mathbf{z})\big]$ is the average reconstruction error, this formulation allows for a better control of the quality of generated data. In the resulting Lagrangian

$$\mathcal{L}(\theta,\phi;\lambda) \equiv \mathbb{E}_{p_\mathcal{D}(\mathbf{x})}\Big[\mathbb{KL}\big(q_\phi(\mathbf{z}|\mathbf{x})\|\, p(\mathbf{z})\big) + \lambda\big(\mathbb{E}_{q_\phi(\mathbf{z}|\mathbf{x})}\big[\mathrm{C}_\theta(\mathbf{x},\mathbf{z})\big] - \kappa^2\big)\Big], \tag{4}$$

the Lagrange multiplier $\lambda$ can be viewed as a weighting term for $\mathbb{E}_{p_\mathcal{D}(\mathbf{x})}\mathbb{E}_{q_\phi(\mathbf{z}|\mathbf{x})}\big[-\log p_\theta(\mathbf{x}|\mathbf{z})\big]$. This approach leads to a similar optimisation objective as in [14] with $\beta = 1/\lambda$. The authors propose a descent-ascent algorithm (GECO) for finding the saddle point of the Lagrangian. The parameters $(\theta,\phi)$ are optimised through gradient descent and $\lambda$ is updated as

$$\lambda_t = \lambda_{t-1} \cdot \exp\big(\nu \cdot (\hat{\mathrm{C}}_t - \kappa^2)\big), \tag{5}$$

corresponding to a quasi-gradient ascent due to $\Delta\lambda_t \cdot \partial_\lambda \mathcal{L} \geq 0$; $\nu$ is the update's learning rate. In the context of stochastic batch gradient training, $\hat{\mathrm{C}}_t \approx \mathbb{E}_{p_\mathcal{D}(\mathbf{x})}\mathbb{E}_{q_\phi(\mathbf{z}|\mathbf{x})}\big[\mathrm{C}_\theta(\mathbf{x},\mathbf{z})\big]$ is estimated as the running average $\hat{\mathrm{C}}_t = (1-\alpha)\cdot\hat{\mathrm{C}}_{\text{ba}} + \alpha\cdot\hat{\mathrm{C}}_{t-1}$, where $\hat{\mathrm{C}}_{\text{ba}}$ is the batch average $\mathbb{E}_{p_\mathcal{D}(\mathbf{x}_{\text{ba}})}\mathbb{E}_{q_\phi(\mathbf{z}|\mathbf{x})}\big[\mathrm{C}_\theta(\mathbf{x},\mathbf{z})\big]$. To the best of our understanding,[1] the GECO algorithm solves the optimisation problem

$$\min_\theta \max_\lambda \min_\phi \mathcal{L}(\theta,\phi;\lambda) \quad \text{s.t.} \quad \lambda \geq 0. \tag{6}$$

Here, $\max_\lambda \min_\phi \mathcal{L}(\theta,\phi;\lambda)$ can be viewed to correspond to the E-step of the EM algorithm. However, in general this objective can only be guaranteed to be the ELBO if $\lambda = 1$, or in case of $0 \leq \lambda < 1$, a scaled lower bound on the ELBO.

## 2.2 Hierarchical Priors for Learning Informative Latent Representations

In this section, we propose a hierarchical prior for VAEs within the constrained optimisation setting. Our goal is to incentivise the learning of informative latent representations and to avoid over-regularising the posterior distribution (i) by increasing the complexity of the prior distribution $p(\mathbf{z})$, and (ii) by providing an optimisation method to learn such models.

It has been shown in [28] that the optimal empirical Bayes prior is the aggregated posterior distribution $p^*(\mathbf{z}) = \mathbb{E}_{p_{\mathcal{D}}(\mathbf{x})}\big[q_\phi(\mathbf{z}|\mathbf{x})\big]$. We follow [28] to approximate this distribution in the form of a mixture distribution. However, we opt for a continuous mixture/hierarchical model

$$p_\Theta(\mathbf{z}) = \int p_\Theta(\mathbf{z}|\zeta)\, p(\zeta)\, \mathrm{d}\zeta, \tag{7}$$

with a standard normal $p(\zeta)$. This leads to a hierarchical model with two stochastic layers. As a result, intuitively, our approach inherently favours the learning of continuous latent features. We refer to this model by variational hierarchical prior (VHP).

In order to learn the parameters in Eq. (7), we propose to adapt the constrained optimisation problem in Sec. 2.1 to hierarchical models. For this purpose we use an importance-weighted (IW) bound [5]—and the corresponding proposal distribution $q_\Phi$—to introduce a sequence of upper bounds

$$\mathbb{E}_{p_{\mathcal{D}}(\mathbf{x})} \mathbb{KL}\big(q_\phi(\mathbf{z}|\mathbf{x})\| p(\mathbf{z})\big) \leq \mathcal{F}(\phi, \Theta, \Phi)$$

$$\equiv \mathbb{E}_{p_{\mathcal{D}}(\mathbf{x})} \mathbb{E}_{q_\phi(\mathbf{z}|\mathbf{x})} \left[ \log q_\phi(\mathbf{z}|\mathbf{x}) - \mathbb{E}_{\zeta_{1:K} \sim q_\Phi(\zeta|\mathbf{z})} \left[ \log \frac{1}{K} \sum_{k=1}^{K} \frac{p_\Theta(\mathbf{z}|\zeta_k)\, p(\zeta_k)}{q_\Phi(\zeta_k|\mathbf{z})} \right] \right], \tag{8}$$

with $K$ importance weights, resulting in an upper bound on Eq. (4):

$$\mathcal{L}(\theta, \phi; \lambda) \leq \mathcal{F}(\phi, \Theta, \Phi) + \lambda\big(\mathbb{E}_{p_{\mathcal{D}}(\mathbf{x})} \mathbb{E}_{q_\phi(\mathbf{z}|\mathbf{x})}\big[C_\theta(\mathbf{x},\mathbf{z})\big] - \kappa^2\big) \equiv \mathcal{L}_{\text{VHP}}(\theta, \phi, \Theta, \Phi; \lambda). \tag{9}$$

As a result, we arrive to the optimisation problem

$$\min_{\Theta,\Phi} \min_\theta \max_\lambda \min_\phi \mathcal{L}_{\text{VHP}}(\theta, \phi, \Theta, \Phi; \lambda) \quad \text{s.t.} \quad \lambda \geq 0, \tag{10}$$

which we can optimise by the following double-loop algorithm: (i) in the *outer loop* we update the bound w.r.t. $(\Theta, \Phi)$; (ii) in the *inner loop* we solve the optimisation problem $\min_\theta \max_\lambda \min_\phi \mathcal{L}_{\text{VHP}}(\theta, \phi, \Theta, \Phi; \lambda)$ by applying an update scheme for $\lambda$ and $\beta = 1/\lambda$, respectively. In the following, we use the $\beta$-parameterisation to be in line with [e.g. 14, 27].

In the GECO update scheme (Eq. (5)), $\beta$ increases/decreases until $\hat{C}_t = \kappa^2$. However, provided the constraint is fulfilled, we want to obtain a tight lower bound on the log-likelihood. As discussed in Sec. 2.1, this holds when $\beta = 1$ (ELBO)—in case of $\beta > 1$, we would optimise a scaled lower bound on the ELBO. Therefore, we propose to replace the corresponding $\beta$-version of Eq. (5) by

$$\beta_t = \beta_{t-1} \cdot \exp\big[\nu \cdot f_\beta(\beta_{t-1}, \hat{C}_t - \kappa^2; \tau) \cdot (\hat{C}_t - \kappa^2)\big], \tag{11}$$

where we define

$$f_\beta(\beta, \delta; \tau) = \big(1 - H(\delta)\big) \cdot \tanh\big(\tau \cdot (\beta - 1)\big) - H(\delta). \tag{12}$$

Here, $H(\cdot)$ is the Heaviside function and we introduce a slope parameter $\tau$. This update can be interpreted as follows. If the constraint is violated, i.e. $\hat{C}_t > \kappa^2$, the update scheme is equal to Eq. (5). In case the constraint is fulfilled, the $\tanh$ term guarantees that we finish at $\beta = 1$, to obtain/optimise the ELBO at the end of the training. Thus, we impose $\beta \in (0, 1]$, which is reasonable since $\beta < \beta_{\max}$ does not violate the constraint. A visualisation of the $\beta$-update scheme is shown in Fig. 1. Note that there are alternative ways to modify Eq. (5), see App. B.1, however, Eq. (11) led to the best results.

The double-loop approach in Eq. (10) is often computationally inefficient. Thus, we decided to run the inner loop only until the constraints are satisfied and then updating the bound. That is, we optimise Eq. (10) and skip the outer loop/bound updates when the constraints are not satisfied. It turned out that the bound updates were often skipped in the initial phase, but rarely skipped later on. Hence, the algorithm behaves as layer-wise pretraining [3]. For these reasons, we propose Alg. 1 (REWO) that separates training into two phases: an initial phase, where we only optimise the reconstruction error—and a main phase, where all parameters are updated jointly.

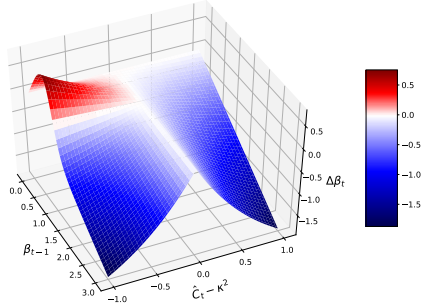

Figure 1: $\beta$-update scheme: $\Delta\beta_t = \beta_t - \beta_{t-1}$ as a function of $\beta_{t-1}$ and $\hat{C}_t - \kappa^2$ for $\nu = 1$ and $\tau = 3$. A comparison with the GECO update scheme can be found in App. A. (see Sec. 2.2)

---

**Algorithm 1** (REWO) Reconstruction-error-based weighting of the objective function

---

Initialise $t = 1$
Initialise $\beta \ll 1$
Initialise INITIALPHASE = TRUE
**while** training **do**
    Read current data batch $\mathbf{x}_{\text{ba}}$
    Sample from variational posterior $\mathbf{z} \sim q_\phi(\mathbf{z}|\mathbf{x}_{\text{ba}})$
    Compute $\hat{C}_{\text{ba}}$ (batch average)
    $\hat{C}_t = (1 - \alpha) \cdot \hat{C}_{\text{ba}} + \alpha \cdot \hat{C}_{t-1}, \quad (\hat{C}_0 = \hat{C}_{\text{ba}})$
    **if** $\hat{C}_t < \kappa^2$ **then**
        INITIALPHASE = FALSE
    **end if**
    **if** INITIALPHASE **then**
        Optimise $\mathcal{L}_{\text{VHP}}(\theta, \phi, \Theta, \Phi; \beta)$ w.r.t $\theta, \phi$
    **else**
        $\beta \leftarrow \beta \cdot \exp\left[\nu \cdot f_\beta(\beta_{t-1}, \hat{C}_t - \kappa^2; \tau) \cdot (\hat{C}_t - \kappa^2)\right]$
        Optimise $\mathcal{L}_{\text{VHP}}(\theta, \phi, \Theta, \Phi; \beta)$ w.r.t $\theta, \phi, \Theta, \Phi$
    **end if**
    $t \leftarrow t + 1$
**end while**

---

In the initial phase, we start with $\beta \ll 1$ to enforce a reconstruction optimisation. Thus, we train the first stochastic layer for achieving a good encoding of the data through $q_\phi(\mathbf{z}|\mathbf{x})$, measured by the reconstruction error. For preventing $\beta$ to become smaller than the initial value during the first iteration steps, we start to update $\beta$ when the condition $\hat{C}_t < \kappa^2$ is fulfilled. A good encoding is required to learn the conditionals $q_\Phi(\zeta|\mathbf{z})$ and $p_\Theta(\mathbf{z}|\zeta)$ in the second stochastic layer. Otherwise, $q_\Phi(\zeta|\mathbf{z})$ would be strongly regularised towards $p(\zeta)$, resulting in $\mathbb{KL}\left(q_\Phi(\zeta_i|\mathbf{z})\| p(\zeta_i)\right) \approx 0$, from which it typically does not recover [27]. In the main phase, after $\hat{C}_t < \kappa^2$ is fulfilled, we start to optimise the parameters of the second stochastic layer and to update $\beta$. This approach avoids posterior collapse in both stochastic layers (see Sec. 4.1 and App. D.2), and thereby helps the prior to learn an informative encoding for preventing the aforementioned over-regularisation.

The proposed method, which is a combination of an ELBO-like Lagrangian and an IW bound, can be interpreted as follows: the posterior of the first stochastic layer $q_\phi(\mathbf{z}|\mathbf{x})$ can learn an informative latent representation due to the flexible hierarchical prior. The flexible prior, on the other hand, is achieved by applying an IW bound. Despite a diagonal Gaussian $q_\Phi(\zeta|\mathbf{z})$, the importance weighting allows to learn a precise conditional $p_\Theta(\mathbf{z}|\zeta)$ from the standard normal distribution $p(\zeta)$ to the aggregated posterior $\mathbb{E}_{p_\mathcal{D}(\mathbf{x})}[q_\phi(\mathbf{z}|\mathbf{x})]$ [11]. Alternatively, one could use, for example, a normalising flow [23]. Otherwise, the model could compensate a less expressive prior by regularising $q_\phi(\mathbf{z}|\mathbf{x})$, which would result in a restricted latent representation (see App. B.4 for empirical evidence).

## 2.3 Graph-Based Interpolations for Verifying Latent Representations

A key reason for introducing hierarchical priors was to facilitate an informative latent representation due to less over-regularisation of the posterior. To verify the quality of the latent representations, we build on the manifold hypothesis, defined in [6, 26]. The idea can be summarised by the following assumption: real-world data presented in high-dimensional spaces is likely to concentrate in the vicinity of nonlinear sub-manifolds of much lower dimensionality. Following this hypothesis, the quality of latent representations can be evaluated by interpolating between data points along the learned data manifold in the latent space—and reconstructing this path to the observable space.

To implement the above idea, we propose a graph-based method [8] which summarises the continuous latent space by a graph consisting of a finite number of nodes. The nodes $\mathbf{Z} = \{\mathbf{z}_1, \dots, \mathbf{z}_N\}$ can be obtained by randomly sampling $N$ samples from the learned prior (Eq. (7)):

$$\mathbf{z}_n, \zeta_n \sim p_\Theta(\mathbf{z}|\zeta)\, p(\zeta), \quad n = 1, \dots, N. \tag{13}$$

The graph is constructed by connecting each node by undirected edges to its k-nearest neighbours. The edge weights are Euclidean distances in the latent space between the related node pairs. Once the

graph is built, interpolation between two data points $\mathbf{x}_i$ and $\mathbf{x}_j$ can be done as follows. We encode these data points as $\mathbf{z}_{(\cdot)} = \mu_\theta(\mathbf{x}_{(\cdot)})$, where $\mu_\phi(\mathbf{x}_{(\cdot)})$ is the mean of $q_\phi(\mathbf{z}|\mathbf{x}_{(\cdot)})$, and add them as new nodes to the existing graph.

To find the shortest path through the graph between nodes $\mathbf{z}_i$ and $\mathbf{z}_j$, a classic search algorithm such as $A^\star$ can be used. The result is a sequence $\mathbf{Z}_{\text{path}} = \big(\mathbf{z}_i, \mathbf{Z}_{\text{sub}}, \mathbf{z}_j\big)$, where $\mathbf{Z}_{\text{sub}} \subseteq \mathbf{Z}$, representing the shortest path in the latent space along the learned latent manifold. Finally, to obtain the interpolation, we reconstruct $\mathbf{Z}_{\text{path}}$ to the observable space.

## 3   Related Work

Several works improve the VAE by learning more complex priors such as the stick-breaking prior [20], a nested Chinese Restaurant Process prior [13], Gaussian mixture priors [12], or autoregressive priors [10]. A closely related line of research is based on the insight that the optimal prior is the aggregated posterior [28, 19]. The VampPrior [28] approximates the prior by a uniform mixture of approximate posterior distributions, evaluated at a few learned pseudo data points. In our approach, the prior is approximated by using a second stochastic layer (IW bound). The authors in [19] use a two-level stochastic model with a combination of implicit and explicit distributions for the encoders and decoders. Inference is done through optimising a sandwich bound of the ELBO, which is specific to the choice of implicit distributions. In our work, however, we address inference using a constrained optimisation approach and all distributions are explicit.

In the context of VAEs, hierarchical latent variable models were already introduced earlier [24, 5, 27]. Compared to our approach, these works have in common the structure of the generative model but differ in the definition of the inference models and in the optimisation procedure. In our proposed method, the VAE objective is reformulated as the Lagrangian of a constrained optimisation problem. The prior of this ELBO-like Lagrangian is approximated by an IW bound. It is motivated by the fact that a single stochastic layer with a flexible prior can be sufficient for modelling an informative latent representation. The second stochastic layer is required to learn a sufficiently flexible prior.

The common problem of failing to use the full capacity of the model in VAEs [5] has been addressed by applying *annealing/warm-up* [4, 27]. Here, the KL divergence between the approximate posterior and the prior is multiplied by a weighting factor, which is linearly increased from 0 to 1 during training. However, such predefined schedules might be suboptimal. By reformulating the objective as a constrained optimisation problem [25], the above weighting term can be represented by a Lagrange multiplier and updated based on the reconstruction error. Our proposed algorithm builds [25], providing several modifications discussed in Sec. 2.2.

In [14], the authors propose a constrained optimisation framework, where the optimisation objective is the expected negative log-likelihood and the constraint is imposed in the KL term—recall that in [25] the roles are reversed. Instead of optimising the resulting Lagrangian, the authors choose Lagrange multipliers ($\beta$ parameter) that maximise a heuristic cost for disentanglement. Their goal is not to learn a latent representation that reflects the topology of the data but a disentangled representation, where the dimensions of the latent space correspond to various features of the data.

## 4   Experiments

To validate our approach, we consider the following experiments. In Sec. 4.1, we demonstrate that our method learns to represent the degree of freedom in the data of a moving pendulum. In Sec. 4.2, we generate human movements based on the learned latent representations of real-world data (CMU Graphics Lab Motion Capture Database). In Sec. 4.3, the marginal log-likelihood on standard datasets such as MNIST, Fashion-MNIST, and OMNIGLOT is evaluated. In Sec. 4.4, we compare our method on the high-dimensional image datasets 3D Faces and 3D Chairs. The model architectures used in our experiments can be found in App. F.

### 4.1   Artificial Pendulum Dataset

We created a dataset of 15,000 images of a moving pendulum (see Fig. 4). Each image has a size of $16 \times 16$ pixels and the joint angles are distributed uniformly in the range $[0, 2\pi)$. Thus, the joint angle is the only degree of freedom.

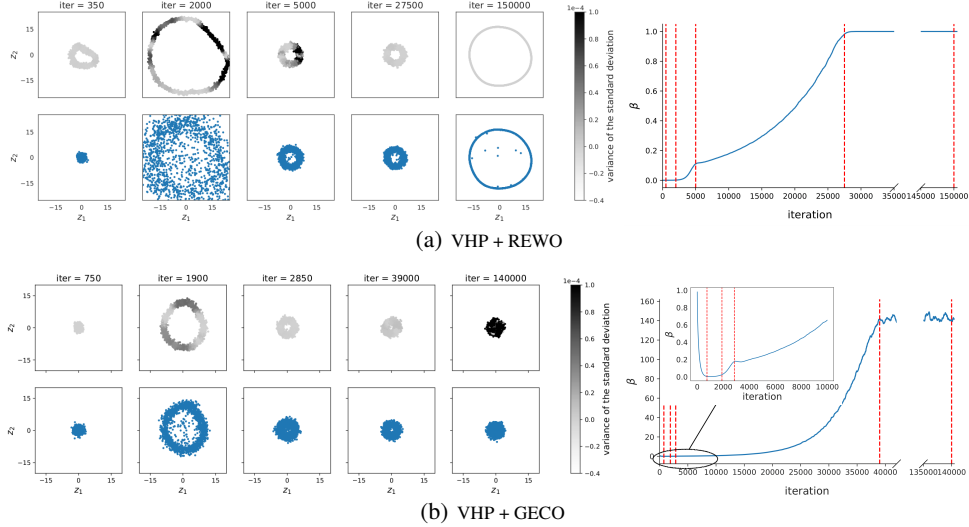

(a) VHP + REWO

(b) VHP + GECO

Figure 2: (left) Latent representation of the pendulum data at different iteration steps when optimising $\mathcal{L}_{\mathrm{VHP}}(\theta, \phi, \Theta, \Phi; \beta)$ with REWO and GECO, respectively. The top row shows the approximate posterior; the greyscale encodes the variance of its standard deviation. The bottom row shows the hierarchical prior. (right) $\beta$ as a function of the iteration steps; red lines mark the visualised iteration steps. (see Sec. 4.1)

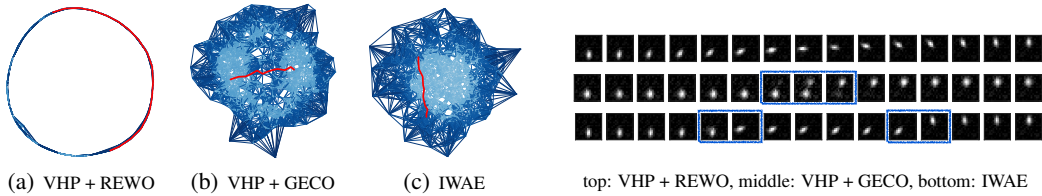

(a) VHP + REWO  (b) VHP + GECO  (c) IWAE

top: VHP + REWO, middle: VHP + GECO, bottom: IWAE

Figure 3: Graph-based interpolation of the pendulum movement. The graph is based on the prior, shown in App. B.5. The red curves depict the interpolations, the bluescale indicates the edge weight. (see Sec. 4.1)

Figure 4: Pendulum reconstructions of the graph-based interpolation in the latent space, shown in Fig. 3. Discontinuities are marked by blue boxes. (see Sec. 4.1)

Fig. 2 shows latent representations of the pendulum data learned by the VHP when applying REWO and GECO, respectively; the same $\kappa$ is used in both cases. The variance of the posterior's standard deviation, expressed by the greyscale, measures whether the contribution to the ELBO is equally distributed over all data points.

To validate whether the obtained latent representation is informative, we apply a linear regression after transforming the latent space to polar coordinates. The goal is to predict the joint angle of the pendulum. We compare REWO with GECO, and additionally with *warm-up* (WU) [27], a linear annealing schedule of $\beta$. In the latter, we use a VAE objective—defined as an ELBO/IW bound combination, similar to Eq. (9); the related plots are in App. B.2. Tab. 1 shows the absolute errors (OLS regression) for the different optimisation procedures; details on the regression can be found in App. B.3. REWO leads to the most precise prediction of the ground truth.

Table 1: OLS regression on the learned latent representations of the pendulum data.

| METHOD | ABSOLUTE ERROR | METHOD | ABSOLUTE ERROR |
|---|---|---|---|
| VHP + REWO | 0.054 | VHP$^\star$ | 0.49 |
| VHP + GECO | 0.53 | VHP$^\star$ + WU (20 EPOCHS) | 0.20 |
| | | VHP$^\star$ + WU (200 EPOCHS) | 0.31 |

$^\star$VAE OBJECTIVE

Furthermore, we demonstrate in App. B.4 that a less expressive posterior $q_\Phi(\zeta|\mathbf{z})$ in the second stochastic layer leads to poor latent representations, since the model compensates it by restricting $q_\phi(\mathbf{z}|\mathbf{x})$—as discussed in Sec. 2.2.

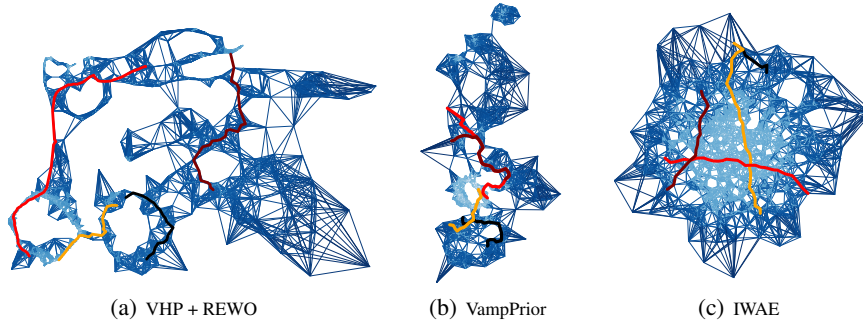

| (a)  VHP + REWO | (b)  VampPrior | (c)  IWAE |

Figure 5: Graph-based interpolation of human motions. The graphs are based on the (learned) prior distributions, depicted in App. C.1. The bluescale indicates the edge weight. The coloured lines represent four interpolated movements, which can be found in Fig. 6 and App. C. (see Sec. 4.2)

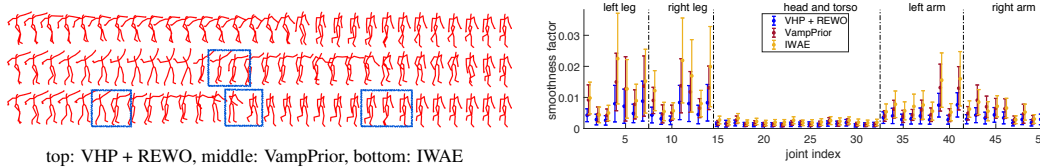

top: VHP + REWO, middle: VampPrior, bottom: IWAE

Figure 6: Human-movement reconstructions of the graph-based interpolations in Fig. 5 (red curve). Reconstruction of the remaining interpolations can be found in App. C.2. Discontinuities are marked by blue boxes. (see Sec. 4.2)

Figure 7: Smoothness measure of the human-movement interpolations. For each joint, the mean and standard deviation of the smoothness factor are displayed. Smaller values correspond to smoother movements. (see Sec. 4.2)

Finally, we compare the latent representations, learned by the VHP and the IWAE, using the graph-based interpolation method. The graphs, shown in Fig. 3, are built (see Sec. 2.3) based on 1000 samples from the prior of the respective model. The red curves depict the interpolation along resulting data manifold, between pendulum images with joint angles of 0 and 180 degrees, respectively. The reconstructions of the interpolations are shown in (Fig. 4). The top row (VHP + REWO) shows a smooth change of the joint angles, whereas the middle (VHP + GECO) and bottom row (IWAE) contain discontinuities resulting in an unrealistic interpolation.

## 4.2   Human Motion Capture Database

The CMU Graphics Lab Motion Capture Database (http://mocap.cs.cmu.edu) consists of several human motion recordings. Our experiments base on five different motions. We preprocess the data as in [7], such that each frame is represented by a 50-dimensional feature vector.

We compare our method with the VampPrior and the IWAE. The prior and approximate posterior of the three methods are depicted in App. C.1. We generate four interpolations (Fig. 5) using our graph-based approach: between two frames within the same motion (black line) and of different motions (orange, red, and maroon); the reconstructions are shown in Fig. 6 and App. C.2. In contrast to the IWAE, the VampPrior and the VHP enable smooth interpolations.

Fig. 7 depicts the movement smoothness factor, which we define as the RMS of the second order finite difference along the interpolated path. Thus, smaller values correspond to smoother movements. For each of the three methods, it is averaged across 10 graphs, each with 100 interpolations. The starting and ending points are randomly selected. As a result, the latent representation learned by the VHP leads to smoother movement interpolations than in case of the VampPrior and the IWAE.

## 4.3   Evaluation on MNIST, Fashion-MNIST, and OMNIGLOT

We compare our method quantitatively with the VampPrior and the IWAE on MNIST [18, 17], Fashion-MNIST [29], and OMNIGLOT [16]. For this purpose, we report the marginal log-likelihood (LL) on the respective test set. Following the test protocol of previous work [28], we evaluate the LL using importance sampling with 5,000 samples [5]. The results are reported in Tab. 2.

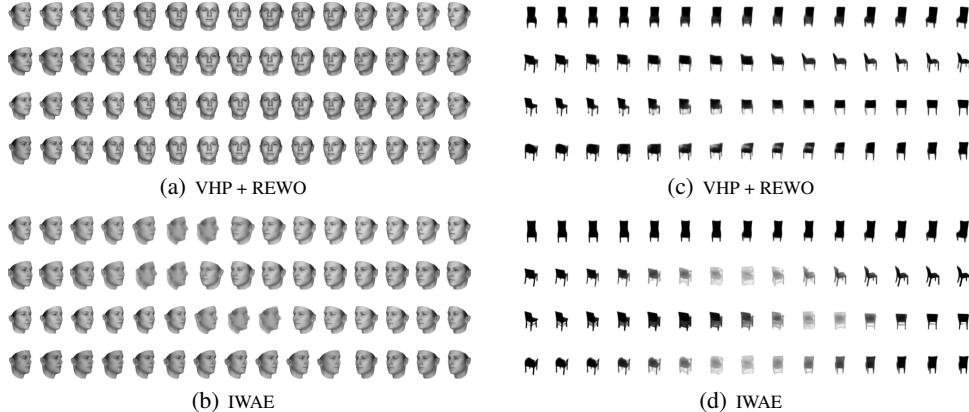

|              |              |              |              |
| :----------: | :----------: | :----------: | :----------: |
| (a) VHP + REWO |            | (c) VHP + REWO |            |
| (b) IWAE     |              | (d) IWAE     |              |

Figure 8: Faces & Chairs: graph-based interpolations—between data points from the test set—along the learned 32-dimensional latent manifold. The graph is based on prior samples. (see Sec. 4.4)

VHP + REWO performs as good or better than state-of-the-art on these datasets. The same $\kappa$ was used for training VHP with REWO and GECO. The two stochastic layer hierarchical IWAE does not perform better than VHP + REWO, supporting our claim that a flexible prior in the first stochastic layer and a flexible approximate posterior in the second stochastic layer is sufficient. Additionally, we show that REWO leads to a similar amount of active units as WU (see App. D.2).

Table 2: Negative test log-likelihood estimated with 5,000 importance samples.

|              | DYNAMIC MNIST | STATIC MNIST | FASHION-MNIST | OMNIGLOT |
| ------------ | :-----------: | :----------: | :-----------: | :------: |
| VHP + REWO   | 78.88         | 82.74        | 225.37        | 101.78   |
| VHP + GECO   | 95.01         | 96.32        | 234.73        | 108.97   |
| VAMPPRIOR    | 80.42         | 84.02        | 232.78        | 101.97   |
| IWAE (L=1)   | 81.36         | 84.46        | 226.83        | 101.57   |
| IWAE (L=2)   | 80.66         | 82.83        | 225.39        | 101.83   |

## 4.4 Qualitative Results: 3D Chairs and 3D Faces

We generated 3D Faces [22] based on images of 2000 faces with 37 views each. 3D Chairs [2] consists of 1393 chair images with 62 views each. The images have a size of $64 \times 64$ pixels.

Here, our approach is compared with the IWAE using a 32-dimensional latent space. The learned encodings are evaluated qualitatively by using the graph-based interpolation method. Fig. 8(a) and 8(c) show interpolations along the latent manifold learned by the VHP + REWO. Compared to the IWAE (Fig. 8(b) and 8(d)), they are less blurry and smoother. Further results can be found in App. E.

## 5 Conclusion

In this paper, we have proposed a hierarchical prior in the context of variational autoencoders and extended the constrained optimisation framework in *Taming VAEs* to hierarchical models by using an importance-weighted bound on the marginal of the hierarchical prior. Concurrently, we have introduced the associated optimisation algorithm to facilitate good encodings.

We have shown that the learned hierarchical prior is indeed non-trivial, moreover, it is well-adapted to the latent representation, reflecting the topology of the encoded data manifold. Our method provides informative latent representations and performs particularly well on data where the relevant features change continuously. In case of the pendulum (Sec. 4.1), the prior has learned to represent the degrees of freedom in the data—allowing us to predict the pendulum's angle by a simple OLS regression. The experiments on the CMU human motion data (Sec. 4.2) and on the high-dimensional Faces and Chairs datasets (Sec. 4.4) have demonstrated that the learned hierarchical prior leads to smoother and more realistic interpolations than a standard normal prior (or the VampPrior). Moreover, we have obtained test log-likelihoods (Sec. 4.3) comparable to state-of-the-art on standard datasets.

## Acknowledgements

We would like to thank Maximilian Soelch for valuable suggestions and discussions.

## Footnotes

[1]The optimisation problem is not explicitly stated in [25].

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
