[Supplementary Material]

# Appendix

## A  Comparison: $\beta$-Update Scheme in REWO and in GECO

(a) REWO                                     (b) GECO

Figure 9: $\beta$-update scheme: $\Delta\beta_t = \beta_t - \beta_{t-1}$ as a function of $\beta_{t-1}$ and $\hat{C}_t - \kappa^2$ for $\nu = 1$ and $\tau = 3$.

## B  Pendulum

### B.1  Training Process with Alternative $\beta$-Update Scheme

An alternative way to define the $\beta$-update scheme such that $\beta \leq 1\,(\lambda \geq 1)$ is to use Eq. (5) with

$$\lambda_t = 1 + \gamma_t, \quad \text{where} \quad \gamma_t = \gamma_{t-1} \cdot \exp\left(\nu \cdot (\hat{C}_t - \kappa^2)\right). \tag{14}$$

As in Sec. 2, $\nu$ is defined as the update's learning rate. This leads to the following $\beta$-update scheme:

$$\beta_t = \frac{1}{1 + \tau \cdot \gamma_t}, \tag{15}$$

where $\tau$ is a slope parameter. However, the $\beta$-update defined in Eq. (11) is easier to tune, leading to better results. Furthermore, Eq. (11) allows to choose any $\beta > 0$ as starting value.

Figure 10: VHP + REWO (with different $\beta$-update scheme): (left) latent representation of the pendulum data at different iteration steps when optimising $\mathcal{L}_{\mathrm{VHP}}(\theta, \phi, \Theta, \Phi; \beta)$. The top row shows the approximate posterior, where the colour encodes the rotation angle of the pendulum. The bottom row shows samples from the hierarchical prior. (right) $\beta$ as a function of the iteration steps; the red lines mark the visualised iteration steps.

## B.2  Training Process with/without WU

Figure 11: VHP (no REWO/GECO/WU): latent representation of the pendulum data at different iteration steps when optimising $\mathcal{L}_{\mathrm{VHP}}(\theta, \phi, \Theta, \Phi; \beta = 1)$. The top row shows the approximate posterior, where the colour encodes the rotation angle of the pendulum. The bottom row shows samples from the hierarchical prior. It took 27,500 iterations until the model learned a representation of the data. However, the latent representation is less informative than in Fig. 2(a).

Figure 12: VHP + WU (20 epochs): latent representation of the pendulum data at different iteration steps when optimising $\mathcal{L}_{\mathrm{VHP}}(\theta, \phi, \Theta, \Phi; \beta)$. The top row shows the approximate posterior, where the colour encodes the rotation angle of the pendulum. The bottom row shows samples from the hierarchical prior. The model started to learn a representation (iter=2000) but the fast increase $\beta$ led to an over-regularisation by the KL term, resulting in a less informative representation than in Fig. 2(a).

Figure 13: VHP + WU (200 epochs): latent representation of the pendulum data at different iteration steps when optimising $\mathcal{L}_{\mathrm{VHP}}(\theta, \phi, \Theta, \Phi; \beta)$. The top row shows the approximate posterior, where the colour encodes the rotation angle of the pendulum. The bottom row shows samples from the hierarchical prior. The learned latent representation less informative than in Fig. 2(a).

### B.3 OLS Regression on Learned Latent Representations

Fig. 14 shows the joint angle versus $\arcsin(z_2/r)$, where $z_2$ is the second component of the latent space and the radius $r$ is estimated from the learned latent representation.

Figure 14: Verifying the learned latent representations of the VHP trained with REWO, GECO, or WU: OLS regressions on encodings of the pendulum data. The absolute errors are shown in Tab. 1.

### B.4 VHP with ELBO instead of IW Bound

Figure 15: VHP + REWO (with ELBO instead of IW bound in the second stochastic layer): (left) latent representation of the pendulum data at different iteration steps when optimising $\mathcal{L}_{\mathrm{VHP}}(\theta, \phi, \Theta, \Phi; \beta)$. The top row shows the approximate posterior, where the colour encodes the rotation angle of the pendulum. The bottom row shows samples from the hierarchical prior. (right) $\beta$ as a function of the iteration steps; the red lines mark the visualised iteration steps. The model compensates the less expressive posterior $q_\Phi(\zeta|\mathbf{z})$ in the second stochastic layer by restricting $q_\phi(\mathbf{z}|\mathbf{x})$, which leads to poor latent representations.

### B.5 Latent Representations Learned by VHP and IWAE

Figure 16: Latent respresentation of VHP + REWO (left), VHP + GECO (middle), and IWAE (right): approximate posterior (top) and prior (bottom). The colour encodes the rotation angle of the pendulum.

# C CMU Human Motion

## C.1 Latent Representations Learned by VHP and IWAE

The prior and aggregated approximate posterior of the three methods are shown in Fig. 17. As expected, for both the VHP and VampPrior the latent representations of different movements are separated. In both cases the learned prior matches the aggregated posterior. By contrast, the IWAE is restricted by the Gaussian prior and cannot represent the different motions separately in the latent space.

Figure 17: Latent representation of human motion data: VHP + REWO (left), VampPrior (middle), and IWAE (right); approximate posterior (top) and prior (bottom). The colour encodes the five human motions. The different sample densities are caused by a different amount of data points for each motion.

## C.2 Graph-Based Interpolation

(a) VHP + REWO

(b) VampPrior

(c) IWAE

Figure 18: Movement interpolation. The different colours correspond to Fig. 5. Discontinuities are marked by blue boxes.

# D Quantitative Results

## D.1 Training Efficiency

Figure 19: NLL vs rate vs distortion on static MNIST

## D.2 Active Units

Furthermore, we evaluate whether REWO prevents over-pruning of the latent variables [30]. Following [27], we evaluate $\mathrm{KL}(q_\Phi(\zeta^j|\mathbf{x}) \| p(\zeta^j))$ for different optimisation strategies, where $\prod_j q_\Phi(\zeta^j|\mathbf{x}) = q_\Phi(\zeta|\mathbf{x})$. We show the results for the inner latent variable on several datasets in Fig. 20.

(a) static MNIST      (b) dynamic MNIST      (c) Fashion-MNIST

(d) OMNIGLOT

Figure 20: Expected KL divergence between approximate posterior and prior for REWO algorithm (left) and WU (right). The latent dimensions are sorted by the KL divergence and the histograms are shown on a logarithmic scale.

# E Faces and Chairs

(a) VHP + REWO

(b) IWAE

Figure 21: Faces: interpolations along the learned latent manifold with a latent space of 32 dimensions.

(a) VHP + REWO

(b) IWAE

Figure 22: Chairs: interpolations along the learned latent manifold with a latent space of 32 dimensions.

# F   Model Architectures

Table 3: Model architectures. GatedFC/GatedConv denote pairs of fully-connected/convolutional layers multiplied element-wise, where one of the layers (gate) always uses sigmoid activations.

| DATASET | OPTIMISER | ARCHITECTURE | |
|---|---|---|---|
| PENDULUM | ADAM $1e$-4 | INPUT<br>LATENTS<br>$q_\phi(\mathbf{z}\|\mathbf{x})$<br>$p_\theta(\mathbf{x}\|\mathbf{z})$<br>$q_\Phi(\zeta\|\mathbf{z})$<br>$p_\Theta(\mathbf{z}\|\zeta)$<br>OTHERS<br>GRAPH | 256(FLATTENED $16\times16$)<br>2<br>FC 256, 256, 256, 256. RELU ACTIVATION.<br>FC 256, 256, 256, 256. RELU ACTIVATION. GAUSSIAN.<br>FC 256, 256, 256, 256, RELU ACTIVATION.<br>FC 256, 256, 256, 256, RELU ACTIVATION.<br>$\kappa = 0.02$, $\nu = 5$, $K = 16$.<br>1,000 NODES, 18 NEIGHBOURS. |
| CMU HUMAN | ADAM $1e$-4 | INPUT<br>LATENTS<br>$q_\phi(\mathbf{z}\|\mathbf{x})$<br>$p_\theta(\mathbf{x}\|\mathbf{z})$<br>$q_\Phi(\zeta\|\mathbf{z})$<br>$p_\Theta(\mathbf{z}\|\zeta)$<br>OTHERS<br>GRAPH | 50<br>2<br>FC 256, 256, 256, 256. RELU ACTIVATION.<br>FC 256, 256, 256, 256. RELU ACTIVATION. GAUSSIAN.<br>FC 256, 256, 256, 256, RELU ACTIVATION.<br>FC 256, 256, 256, 256, RELU ACTIVATION.<br>$\kappa = 0.02$, $\nu = 1$, $K = 32$.<br>2,530 NODES, 15 NEIGHBOURS. |
| FACES,<br>CHAIRS | ADAM $5e$-4 | INPUT<br>LATENTS<br>$q_\phi(\mathbf{z}\|\mathbf{x})$<br><br><br>$p_\theta(\mathbf{x}\|\mathbf{z})$<br>$q_\Phi(\zeta\|\mathbf{z})$<br>$p_\Theta(\mathbf{z}\|\zeta)$<br>OTHERS<br>GRAPH | $64\times64\times1$<br>32<br>CONV $32\times5\times5$(STRIDE 2) , $32\times3\times3$(STRIDE 1), $48\times5\times5$(STRIDE 2).<br>$48\times3\times3$(STRIDE 1), $64\times5\times5$(STRIDE 2), $64\times3\times3$(STRIDE 1).<br>$96\times5\times5$(STRIDE 2), $96\times3\times3$(STRIDE 1), FC 256. RELU ACTIVATION<br>DECONV REVERSE OF ENCODER. RELU ACTIVATION. BERNOULLI.<br>FC 256, 256, RELU ACTIVATION.<br>FC 256, 256, RELU ACTIVATION.<br>$\kappa = 0.2$, $\nu = 1$, $K = 16$.<br>10,000 NODES (FACES), 8,637 NODES (CHAIRS), 18 NEIGHBOURS. |
| DYNAMICMNIST,<br>STATICMNIST,<br>FASHION-MNIST,<br>OMNIGLOT | ADAM $5e$-4 | INPUT<br>LATENTS<br>$q_\phi(\mathbf{z}\|\mathbf{x})$<br><br>$p_\theta(\mathbf{x}\|\mathbf{z})$<br><br><br>$q_\Phi(\zeta\|\mathbf{z})$<br>$p_\Theta(\mathbf{z}\|\zeta)$<br>OTHERS<br><br> | $28\times28\times1$<br>32<br>GATEDCONV $32\times7\times7$(STRIDE 1) , $32\times3\times3$(STRIDE 2),<br>$64\times5\times5$(STRIDE 1), $64\times3\times3$(STRIDE 2), $6\times3\times3$(STRIDE 1)<br>GATEDFC 784, GATEDCONV $64\times3\times3$(STRIDE 1),<br>$64\times3\times3$(STRIDE 1), $64\times3\times3$(STRIDE 1), $64\times3\times3$(STRIDE 1).<br>LINEAR ACTIVATION. BERNOULLI.<br>GATEDFC 256, 256, LINEAR ACTIVATION.<br>GATEDFC 256, 256, LINEAR ACTIVATION.<br>$\kappa = 0.18$ (DYNAMICMNIST, STATICMNIST, OMNIGLOT),<br>$\kappa = 0.31$ (FASHION-MNIST),<br>$\nu = 1$, $K = 16$. |