[Reviews · NeurIPS 2019]

Reviewer 1



This a a solid work. The authors proposed a simple hierarchical prior modelling an infinite mixture and improved state-of-the-art density estimation. This can be seen as an extension to both GECO (Rezende et al) and VampPriorVAE (Tomczak et al), which was also properly cited. The paper is well written, very clear and the evaluation collaborates the expected improvement in performance. Some minor comments to add: * Should the optimisation problem be: min KL(q||p) s.t. E[C(x, z)] < K^2 * I wished the authors could have described more of the results of the graph-based interpolations. In what way is the interpolations done with VHP+REWO better than the models compared to? Is there a way to quantify these results? (Path length of A* algorithm?) * It would be good also to see whether the Lagrangian update alone already leads to a good performance, or whether it is actually the hierarchical prior. * In terms of reporting the results it would be better to do multiple runs and report LL mean + standard error

Reviewer 2



This paper discussed how to enhance the existing methods in which designed prior could over regularize the posteriori, so it will try to find a way to learn a complex prior which can learn the latent pattern of data manifold more efficiently. To learn such prior, paper adopted and modified one dual optimization technique and introduced an efficient algorithm on how to update the hierarchical prior and posteriori parameters. The combination of complex priori with the introduced algorithm have learned a posterior which has more informative latent representation and avoids posteriori collapse. In addition, paper introduced a graph search method to interpolate the states and showed how effective algorithm can discover a meaningful posteriori over the experiment section. So we can summarize the contribution of this paper as following - Introduce a hierarchical prior which can avoid over regularization of the posterior while learning latent variables manifold - Adopting and expanding an optimization technique and an algorithm to learn hierarchical prior and hierarchical posterior parameters. Authors used importance weight techniques to model the prior and hierarchical posterior - Defined an interpolation techniques in latent state and demonstrated its success using different experiments In experiments section, authors demonstrated application of their method using synthetic and real data and showed the proposed method outperformed competing algorithms. Appendix contains very useful information about how changing pieces of algorithm or prior will change the outcome and is convincing why such setup has been picked. Quality Motivation, claims and and supporting material in main paper and supplementary material are explained well and I could not find any significant technical issues with the details of claims made in this paper. The quality of experimental results is very good and many possible variation has been experimented and choice of the authors have been justified. Just one point is to show whether equation 9 which is objective function of our optimization is less that marginal log-likelihood of data under certain circumferences for \lambda and \kappa. Clarity: I think the paper objectives and explanation are pretty clear and flow of material is very smooth. Originality: As mentioned in summary the main contribution of this paper could be summarized as bellow - Introduce a hierarchical prior which can avoid over regularization of the posterior while learning latent variables manifold - Adopting and expanding an optimization technique and an algorithm to learn hierarchical prior and hierarchical posterior parameters. Authors used importance weight techniques to model the prior and hierarchical posterior - Defined an interpolation techniques in latent state and demonstrated its success using different experiments The paper has solid original contribution. Authors have done a detailed literature review and most of related works have been mentioned and contribution of this paper have been compared and highlighted clearly. My only suggestion is that authors can take a look at the paper Molchanov, Dmitry, et al. "Doubly semi-implicit variational inference." arXiv preprint arXiv:1810.02789 (2018). which use semi-implicit hierarchical prior and hierarchical posteriori distribution which makes it very similar to this paper. I would belive that optimization of parameters is more efficient than doubly semi-implicit due to how the inner-outer loop has been handled in this paper. Significance:  The method is new and original. I think the experiments section has answered many question that I had while I was reading paper and extend of experiments are higher than average NIPS submission. Just as mentioned in originality section, I would like to refer the authors to Molchanov, Dmitry, et al. "Doubly semi-implicit variational inference." arXiv preprint arXiv:1810.02789 (2018) which they potentially can compare their performance to the method purposed in that paper.

Reviewer 3



The paper contributes an extension to GECO which admits a hierarchical prior p(z) = \int p(zeta)p(z|zeta) \dd{zeta} whose likelihood is intractable. The motivation for such a prior is to combat posterior collapse and enable better disentanglement. The authors then design an importance-weighted upperbound on KL(q(z|x) || p(z)) by lowerbounding log p(z) using samples from an importance distribution q(zeta|z). GECO as well as REWO (the proposed variant) rely on constrained optimisation, where the constraint is replaced by a weighted penalty, whose weight (the Lagrange multiplier lambda) is optimised via SGD. In this paper the authors propose a modification of the update for lambda that promotes convergence to a proper ELBO whenever the constraint is satisfied. The constraint corresponds to a pre-specified expected reconstruction loss. The novelty is in the combination of techniques (GECO + importance weighted bounds) and the different update rule for the Lagrange multiplier. I found the paper quite clear though the argumentation is sometimes a bit too informal (see for example, lines 123--131, where authors list merits of the technique--perhaps justified empirically---which are hard to predict from design choices alone).

[Author Response · NeurIPS 2019]

We thank the reviewers for their invested time and constructive criticism. We believe that their suggestions will
significantly improve the manuscript. In the following, the comments are addressed separately for each reviewer.

**Reviewer-1**

i) "Should the optimisation problem: $\min \mathbb{KL}(q \| p)$ s.t. $\mathbb{E}[C(\mathbf{x}, \mathbf{z})] < \kappa^2$":
Yes, indeed our optimisation objective starts from $\min \mathbb{KL}(q \| p)$ s.t. $\mathbb{E}[C(\mathbf{x}, \mathbf{z})] < \kappa^2$ (line 58). Eq. (6) defines the
corresponding Lagrange dual problem. To extend this to a two level stochastic model, we additionally use an IW upper
bound on $\log p(\mathbf{z})$ inside the KL—Eq. (8)—to accommodate the hierarchical representation of $\log p(\mathbf{z})$. This leads to
the optimisation problem in Eq. (10). We will emphasise this in the manuscript.

ii) "In what way is the interpolations done with VHP+REWO better than...? Is there a way to quantify these results?":
We chose to quantify the graph-based interpolations through the smoothness of the interpolated trajectories in the data
space, as it is one of the desired properties of informative latent representations (Bengio et al.; arXiv:1206.5538). For
this purpose, we introduced a smoothness factor (line 223). When we compare VHP+REWO to the VampPrior and the
standard normal prior, we obtain smoother trajectories (Fig. 7). We will try to expand this part of the paper.

iii) "It would be good also to see whether the Lagrangian update alone already leads to a good performance, or...":
Our preliminary results showed that the update alone does not guarantee a good performance as it still leads to over-
regularisation due to the standard normal prior. Hence, we obtain unrealistic interpolations similar to Fig. 4 (bottom). It
is the combination of both the Lagrange update (REWO) and the VHP that leads to good performance as shown in
Tab. 1. We will point that out more clearly.

iv) "In terms of reporting the results it would be better to do multiple runs and report LL mean + standard error":
We agree on that and we are trying to close this gap. We have been somewhat limited by the number of GPUs we have
access to (depending on the dataset, one optimisation takes over a week on a single GPU until it converges).

**Reviewer-2**

i) "Just one point is to show whether equation 9 which is objective function of our optimization is...":
This is a valid point, we will add "$\log p(\mathcal{D}) \geq \mathcal{L}_{\text{VHP}}(\theta, \phi, \Theta, \Phi; \lambda)$ if $\lambda \geq 1$" before we introduce REWO (line 102).

ii) "My only suggestion is that authors can take a look at the paper Molchanov, Dmitry, et al...":
We thank the reviewer for pointing us to this paper. Indeed, the authors use a similar two-level stochastic model with a
combination of implicit and explicit distributions for the encoder and decoder. Inference is done through optimising
a sandwich bound of the ELBO, which is specific to the choice of implicit distributions. In our work, however, we
address inference using a constrained optimisation approach and our distributions are all explicit. We will definitely cite
and discuss Molchanov, Dmitry, et al. (2018) in the related work.

**Reviewer-3**

i) "I found the paper quite clear though the argumentation is sometimes a bit too informal (see for example, lines 123...":
We thank the reviewer for pointing us to this paragraph. It can indeed be improved in terms of clarity—we will
emphasise that the intuitions in this paragraph are mostly based on empirical evidence.

ii) "...though it would have been nice to see an ablation for REWO itself using priors of different complexity.":
That is an interesting question and we have run some selective experiments, where we combined REWO with the
VampPrior and the standard normal prior. Generally, we observed that: i) REWO alone makes the optimisation less
sensitive to hyperparameters like the network architecture, and ii) it guarantees that the reconstruction is not neglected
in favour of a low KL. However, we decided that experiments in these directions would take the focus from our main
message: learning informative latent representations. On the other hand, if we want to judge REWO only based on the
quality of the latent representation, it is beneficial to use an arbitrary flexible prior (experiments in Tab. 1).

iii) "Similarly, to which extent the modification of the update rule for lambda contributes to results?":
We compared GECO to REWO (modified update rule) on our two-level stochastic model (Sec. 4.1 & 4.3). Apart from
obtaining better ELBO values (Tab. 2) at the end of training, REWO led to more informative latent representations, as
shown in the graph-based interpolations (Fig. 4) and the OLS regression (Tab. 1).

iv) "In Related work you discuss connections with VampPrior which uses the same inference network q(z|x)...":
Thank you for pointing that out, we will emphasise the need of an additional $q(\zeta | \mathbf{z})$ in comparison to the VampPrior.

v) "The text also says (line 158–159) "the aggregated posterior is...". I think a better wording would...":
Yes, this is indeed a more accurate description, we will reword the sentence as suggested—and also replace "aggregated
posterior" by "prior" in the previous sentence (line 156).

[Meta-Review · NeurIPS 2019]

All reviewers are uninamously for accept. The work has clear and interesting leveraging hierarchical priors as an extension of GECO and with importance weighted losses. The literature review and related work is well-written (which the reviewers also agree).